# In Silico Prediction of a Multitope Vaccine against *Moraxella catarrhalis*: Reverse Vaccinology and Immunoinformatics

**DOI:** 10.3390/vaccines9060669

**Published:** 2021-06-18

**Authors:** Mohamed A. Soltan, Nada Elbassiouny, Helmy Gamal, Eslam B. Elkaeed, Refaat A. Eid, Muhammad Alaa Eldeen, Ahmed A. Al-Karmalawy

**Affiliations:** 1Department of Microbiology and Immunology, Faculty of Pharmacy, Sinai University, Ismailia 41611, Egypt; mohamed.mohamed@su.edu.eg; 2Department of Pharmacology, Faculty of Pharmacy, Sinai University, Ismailia 41611, Egypt; nada.elbasuny@su.edu.eg; 3Biochemistry Division, Chemistry Department, Faculty of Science, Mansoura University, Mansura 35516, Egypt; helmy.mohamed@su.edu.eg; 4Department of Pharmaceutical Sciences, College of Pharmacy, AlMaarefa University, Ad Diriyah, Riyadh 13713, Saudi Arabia; ikaeed@mcst.edu.sa; 5Department of Pharmaceutical Organic Chemistry, Faculty of Pharmacy (Boys), Al-Azhar University, Cairo 11884, Egypt; 6Department of Pathology, College of Medicine, King Khalid University, Abha 62529, Saudi Arabia; refaat_eid@yahoo.com; 7Cell Biology, Histology & Genetics Division, Zoology Department, Faculty of Science, Zagazig University, Zagazig 44519, Egypt; dr.muhammadalaa@gmail.com; 8Department of Pharmaceutical Medicinal Chemistry, Faculty of Pharmacy, Horus University-Egypt, New Damietta 34518, Egypt

**Keywords:** *Moraxella catarrhalis*, vaccinomics, reverse vaccinology, immunoinformatics, epitope mapping, multitope vaccine

## Abstract

*Moraxella catarrhalis* (*M. catarrhalis*) is a Gram-negative bacterium that can cause serious respiratory tract infections and middle ear infections in children and adults. *M. catarrhalis* has demonstrated an increasing rate of antibiotic resistance in the last few years, thus development of an effective vaccine is a major health priority. We report here a novel designed multitope vaccine based on the mapped epitopes of the vaccine candidates filtered out of the whole proteome of *M. catarrhalis*. After analysis of 1615 proteins using a reverse vaccinology approach, only two proteins (outer membrane protein assembly factor BamA and LPS assembly protein LptD) were nominated as potential vaccine candidates. These proteins were found to be essential, outer membrane, virulent and non-human homologs with appropriate molecular weight and high antigenicity score. For each protein, cytotoxic T lymphocyte (CTL), helper T lymphocyte (HTL) and B cell lymphocyte (BCL) epitopes were predicted and confirmed to be highly antigenic and cover conserved regions of the proteins. The mapped epitopes constituted the base of the designed multitope vaccine where suitable linkers were added to conjugate them. Additionally, beta defensin adjuvant and pan-HLA DR-binding epitope (PADRE) peptide were also incorporated into the construct to improve the stimulated immune response. The constructed multitope vaccine was analyzed for its physicochemical, structural and immunological characteristics and it was found to be antigenic, soluble, stable, non-allergenic and have a high affinity to its target receptor. Although the in silico analysis of the current study revealed that the designed multitope vaccine has the ability to trigger a specific immune response against *M. catarrhalis*, additional translational research is required to confirm the effectiveness of the designed vaccine.

## 1. Introduction

Generally, *M. catarrhalis* is known to be a human-restricted commensal, but studies in recent decades have proved that it has become a major infectious agent in both the upper and lower respiratory tract and the causative agent of about 17% of acute otitis media infections in children [1]. Furthermore, *M. catarrhalis* was found to cause other types of infections, such as bacteremia, endocarditis and meningitis, especially in immunocompromised patients [2].

Usage of antibiotics without restrictions has contributed largely to the development of resistant bacterial strains which in turn shorten the list of available effective antimicrobial agents [3]. The resistance rates are rapidly increasing, particularly with regard to first-line antimicrobial agents. An alternative solution is the development of an effective vaccine; while several predicted vaccines against *M. catarrhalis* are in different stages of development, there is no FDA-approved one to date [4]. Adhesive proteins of *M. catarrhalis* were extensively targeted as potential vaccine candidates. They are outer membrane proteins that allow the microorganism to bind to its target receptor in the human respiratory tract, which is the first step in bacterial pathogenesis [5]. Application of this vaccine in animal models showed an enhanced lung clearance of bacteria [6]. 

Vaccine development through traditional approaches has the disadvantages of being expensive, time consuming and seldom giving successful outcomes [7]. Recent development in the fields of bioinformatics, immunoinformatics and structural vaccinomics has revolutionized the process of antigen screening. By merging these techniques, a new approach of vaccine development called reverse vaccinology was created [8] and applied for vaccine development against many bacterial and viral pathogens [9]. Recently, it was demonstrated that a multitope vaccine showed superior efficacy and protection against infectious agents, more so than single epitope-based vaccines [10].

In the current study, the reverse vaccinology approach has been applied to filter the complete proteome of *M. catarrhalis* and nominate potential vaccine candidates. This approach was chosen not only because of its advantages, but also because it selects the candidates from extracellular and outer membrane proteins which demonstrated promising results in animal models in previous trials. Additionally, T and B cell epitopes were mapped for the filtered proteins and the top-ranked epitopes were applied to construct a multitope vaccine which was analyzed according to its structural and immunological characteristics to support its nomination as a promising vaccine candidate against *M. catarrhalis*.

## 2. Materials and Methods

The flow of work started with the application of the reverse vaccinology approach to select which proteins would be our vaccine candidates, then B and T cell epitopes were mapped for these candidates and, following that, the multitope vaccine was constructed and, finally, it was analyzed for its physicochemical, structural and immunological characteristics.

### 2.1. Data Retrieval and Proteome Analysis

The whole proteome of *M. catarrhalis* BBH18 was retrieved (GenBank assembly accession no: GCA_000092265.1). This strain was selected as it represents the reference *M. catarrhalis* strain in NCBI. The reverse vaccinology approach was applied to perform protein prioritization on the whole proteome of *M. catarrhalis*. Firstly, essential proteins, without which *M. catarrhalis* cannot survive, were detected using the Geptop 2.0 webserver with an essentiality cutoff of 0.24. This server detects essential proteins through comparing the orthology and phylogeny of query proteins against the datasets defined experimentally in the database of essential genes (DEG) [11]. Following that, the PSORTb v3.0.2 online server [12] was used to analyze the subcellular localization of the essential proteins. The final outcome of this filtration step was the detection of exoproteome and secretome essential proteins. These proteins were applied in VICMpred [13] to estimate the virulence potential. Proteins filtered out from the previous step were searched using BLASTp from NCBI against the human proteome and proteins with ≥35% identity were excluded as they were considered human homologs. The presence of transmembrane helices in the proteins from the previous step was detected by TMHMM [14], while protein molecular weight was predicted on the Expasy webserver [15]. Proteins with a molecular weight of less than 110 kDa and with ≤1 transmembrane helix were moved to the final filtration step of antigenicity detection by Vaxijen V 2.0 [16]. After all the steps of proteome mining, top proteins that covered all the previous requirements and with the highest antigenicity score were nominated as the potential recombinant vaccine candidates of the study and their protein–protein interactions were assessed using the STRING webserver. Finally, they moved to the epitope mapping stage to construct the multitope vaccine.

### 2.2. T and B Cell Epitope Prediction

Before epitope prediction, the amino acid sequence of filtered proteins was submitted to the SignalP-5.0 Server to predict the presence and location of signal peptides. Following that, the obtained mature polypeptides were analyzed using the prediction tools in the Immune Epitope Database (IEBD). MHC-I binding was detected by applying NetMHCpan EL 4.0 as a prediction method and an HLA allele reference set which provides >97% in terms of population coverage [17], while IEBD recommended a 2.22 prediction method and a full HLA reference set that covers >99% in terms of population coverage [18] for analysis of MHC-II binding. Antibody epitope prediction was also performed with IEBD. Finally, the conservation profile of each selected epitope was analyzed through multiple sequence alignment available with Clustal Omega, while allergenicity assessment and toxicity analysis were performed with AllerTOP and ToxinPred servers, respectively.

### 2.3. Multitope Vaccine Construction

The designed vaccine started with a β defensin adjuvant, then top-ranked CTL, HTL and BCL of the nominated proteins were incorporated using GGGS, GPGPG and KK linkers, respectively. These linkers were added to make sure that successful separation in vivo was obtained [19]. The PADRE sequence was also added to our designed vaccine, and before moving to the next step, the constructed vaccine was confirmed to be antigenic and non-allergic using VaxiJen v2.0 and AllerTOP servers, respectively.

### 2.4. Physicochemical Characterization, Protein Solubility Assessment and Secondary Structure Prediction

ProtParam, a tool available on the Expasy server, was used for physicochemical characterization of the constructed vaccine. This tool detects many physicochemical characteristics of uploaded amino acids, such as molecular weight and isoelectric pH, in addition to several physicochemical properties. The propensity of the constructed vaccine upon overexpression in *E. coli* was assessed by SOLpro while the protein secondary structure was predicted on the PSIPRED server.

### 2.5. Tertiary Structure Prediction, Refinement and Validation

Protein tertiary structure prediction servers perform bending and twisting to generate a protein molecule with the lowest energy state and maximum stability. In the current study, 3Dpro [20] was employed for this purpose. 3Dpro performs its 3D modeling through analysis of the structural similarity between the query protein and the data available on PDB. The GalaxyRefine server [21] was used to refine the protein tertiary structure predicted by 3Dpro to enhance its accuracy, where the refined models were assessed by Ramachandran plot analysis [22] and ProSA [23].

### 2.6. Disulfide Engineering of the Designed Vaccine

Before moving to the next step and starting docking analysis, it is important to improve the designed model stability. Disulfide bonds improve protein geometric conformation and significantly increase their stability. Disulfide by Design 2.0 [24] was used to assign such bonds for the designed vaccine.

### 2.7. Docking of Designed Vaccine with TLR2

Molecular docking was applied to investigate the preferred orientation of the ligand to its corresponding receptor and analyze the binding affinity [25]. Inflammation triggered by *M. catarrhalis* mainly involves TLR2 [26]. Hence, TLR2 (PDB id: 2Z7X) was selected as a docking receptor while the refined model of vaccine tertiary structure was used as a ligand. Molecular docking was studied using the ClusPro 2.0 server [27].

### 2.8. Molecular Dynamics Simulation

Molecular dynamics is a computational approach that was applied in order to describe the molecules’ behavior as well as measure the stability of protein–protein complexes [28]. We employed the iMODS server to study and analyze the interaction between the designed vaccine and its receptor as it has the advantage of being fast and efficient [29]. This tool estimates the direction and range of the basic motions of the protein–ligand complex by measuring four main factors: deformability, eigenvalues, B-factors and covariance. Generally, deformation is much harder when the eigenvalue is high [30].

### 2.9. Reverse Translation and Codon Adaptation

*E. coli* k-12 was selected as the expression host of the recombinant vaccine and, to accelerate the rate of protein expression, codon adaptation was performed to adapt the codon to the chosen expression host. There are no similarities between the codon usage of humans and *E. coli* and, consequently, codon adaptation was performed to get a high expression rate of vaccine protein in the nominated host. The JCAT server [31] was employed for this process.

## 3. Results

### 3.1. In Silico Proteome Analysis

The total number of proteins of *M. catarrhalis* BBH18 was found to be 1615, and 395 of them were predicted as essential proteins. Subcellular localization of essential proteins demonstrated that two proteins, nucleoside-diphosphate kinase and tRNA (adenosine(37)-N6)-threonylcarbamoyltransferase complex transferase subunit TsaD, were located extracellularly while three proteins, LPS assembly protein LptD, outer membrane protein assembly factor BamA and outer membrane protein assembly factor BamD, were outer membrane proteins. Virulence assessment of these proteins revealed that only LPS assembly protein LptD and outer membrane protein assembly factor BamA were virulent, hence other protein filtration steps were performed only for these proteins (Figure 1) where analysis results (Table 1) confirmed their nomination as potential vaccine candidates of the current study.

### 3.2. Protein–Protein Interaction (PPI) Analysis

Interactions between protein candidates and other *M. catarrhalis* proteins were demonstrated using the STRING database (Figure 2). LptD and BamA were predicted to interact with a few other *M. catarrhalis* proteins and also each other. LptD was found to interact with other lipopolysaccharide assembly proteins such as LptF and LptA to form the lipopolysaccharide export system, while outer membrane protein assembly factor BamA interacts with BamD to constitute the outer membrane protein assembly complex. Both LptD and BamA were found to interact with other proteins (such as chaperone SurA and lipoprotein yfgL precursor) that are involved in the correct folding and assembly of outer membrane proteins.

### 3.3. T Cell Epitopes

For both MHC-I and MHC-II peptides, the top 1% predicted epitopes for each protein were analyzed according to antigenicity (estimated by VaxiJen v. 2.0), allergenicity (estimated by AllerTOP v. 2.0) and toxicity (estimated by ToxinPred), and the top 10 peptides that had the lowest percentile rank and antigenicity score >0.4 are listed in Table 2 (for MHC-I peptides) and Table 3 (for MHC-II peptides).

### 3.4. Construction of Multitope Vaccine with Physiochemical Property Assessment and Secondary Structure Prediction

The designed vaccine consisted of six CTL epitopes, six HTL epitopes and six BCL epitopes (three epitopes for each protein) linked in a sequential manner. Epitopes were selected from Table 2, Table 3 and Table 4 based on their characteristics, including epitopes that were non-allergic, non-toxic, conserved, covered different spots in the protein sequence and highly antigenic, were chosen for the multitope vaccine construction. In addition to that, the beta defensin adjuvant and PADRE peptide sequence were also added, resulting in a multiepitope vaccine that was 378 amino acids in length and its sequence was as follows.

“EAAAKGIINTLQKYYCRVRGGRCAVLSCLPKEEQIGKCSTRGRCCRRKKEAAAKAKFVAAWTLKAAAGGGSYSAGVGATWGGGSAEINFEGNRLGGGSDSYGGSLSGGGSRPYARLPQLGGGSRSRIQFDHTWGGGSFTASYPLLRGPGPGVDVEYYIDPVHPVYVGPGPGMTNPYFTVNGVSQSLGPGPGALATFGSELILPLPFGPGPGAVSYRYIDKMGRTRFGPGPGDYNLDYVMDSLMGLNGPGPGDANYLSDFNALGVESKKRKTKYDNKNISNYKKDLTVGFGDKTKKLNKKQNDQTKKSYDEDSLADQNIAKKNGRKKHIRVGINDSESYSSRSSKKYRDAFNPHLSPDKKAKFVAAWTLKAAAGGGS”

The final construction was found to be non-allergenic, soluble with a SOLpro SVM score of 0.88 (SOLpro values > 0.5 are considered soluble) and antigenic with a Vaxijen antigenicity score of 1.07. The physiochemical properties of our construct were detected by using the ProtParam online tool (Table 5). Constructed vaccine secondary structure prediction demonstrated the presence of 18% he-lix, 19% strand and 63% coil structure (Figure 3).

### 3.5. Tertiary Structure Prediction, Refinement and Validation

Structure validation of the primary structure, modeled by 3Dpro, showed that 87.5%, 12.2% and 0.3% of residues were located in favored, allowed and disallowed regions, respectively (estimated by Ramachandran plot analysis). In addition to that, the Z-score (estimated by ProSA webserver) of the same model was −3.2. These outcomes revealed that more refinement is required for this initial structure. Protein 3D structure refinement was performed using GalaxyRefine, where the best model (Figure 4A) showed the following enhancement: the Z-score improved from −3.2 to −3.88 (Figure 4B) and the Ramachandran plot analysis improved to show 92.9%, 7.1% and 0% of residues in favored, allowed and outlier regions, respectively (Figure 4C).

### 3.6. Vaccine Disulfide Engineering

Disulfide engineering was performed to stabilize the designed model of the vaccine. Regarding our vaccine, it was found that 29 pairs of amino acids had the capacity to make disulfide bonds by the DbD2 server but after considering other parameters, such as energy and chi3 value, only two pairs were recommended for mutation with cysteine. Hence, a total of four mutations were generated at the residue pairs, namely GLY36–ARG43 and PRO190–LEU202. The accepted values of energy and chi3 were less than 2.2 and −87: +97, respectively.

### 3.7. Molecular Docking of Vaccine with TLR2

ClusPro 2.0 was employed to perform protein–protein molecular docking between the subunit vaccine and TLR2 receptor, where 30 models were created and, among them, we selected the model that occupied the receptor with the best orientation, resulting in high affinity and a low energy score. Model number 2 (Figure 5) fulfilled the selection criteria, where the measured energy score for the selected model was −1352.8, which was lower than the scores of all other predicted docked complexes, demonstrating a stronger affinity.

### 3.8. Molecular Dynamics Simulation

The iMODS server was employed to perform normal mode analysis (NMA) to explore the stability of proteins. The deformability of the complex counts on the individual distortion of each residue, symbolized by hinges in the chain (Figure 6B). The estimated eigenvalue, which represents the motion stiffness of the complex, was 3.38e^−07^ (Figure 6D). Generally, an inverse relationship was found between the eigenvalue and the variance related to each normal mode (Figure 6C). The B-factor scores derived from NMA were equivalent to the RMS (Figure 6A). The covariance matrix indicated the coupling between pairs of residues, where different pairs demonstrated correlated, anti-correlated or uncorrelated motions, represented by red, blue and white, respectively (Figure 6E). An elastic network model was created (Figure 6F), and it revealed the pairs of atoms linked via springs according to the degree of stiffness between them, where stiffer strings appear as darker grays.

### 3.9. Vaccine Reverse Translation and Codon Optimization

The JCat server was employed to perform reverse translation and codon optimization on the amino acid sequence of the multitope vaccineAfter analysis, the measured GC content was 51.58%, which was accepted as it is within the accepted range (30:70%). Moreover, the codon adaptation index (CAI) was found to be 0.93, which demonstrates the high level of expression, as the value of CAI ranges from 0:1 and the accepted range is between 0.8 and 1.

## 4. Discussion

Conventional approaches for vaccine development, such as killed or live attenuated vaccines, are time consuming as it may take decades to develop a successful vaccine. Moreover, many microorganisms are difficult to cultivate or attenuate, resulting in adverse immune responses, which confirms our need for a revolution in the adopted approach for vaccine development [32]. Recently, the process of vaccine development has moved to a less time-consuming and more efficient genome pre-screening-dependent approach [22], and new studies have been based on a novel strategy through adopting immunoinformatics and reverse vaccinology approaches, where the generated subunit vaccine included the antigenic parts only and demonstrated a promising ability for fighting against pathogenic microorganisms [33].

In the current study, we have applied the reverse vaccinology approach on the proteome of *M. catarrhalis* BBH15 to select the most promising proteins as vaccine candidates. Only two proteins (lipopolysaccharide assembly protein LptD and outer membrane protein assembly factor BamA) out of 1615 (the number of all proteins of *M. catarrhalis* BBH15) were found to be essential, outer membrane localized, virulent, antigenic and non-human homologs, with appropriate molecular weight and an accepted number of transmembrane helices (fewer than two) and, therefore, they were selected to be our vaccine candidates. The same approach has been applied successfully to nominate vaccine candidates in bacterial models such as *Staphylococcus aureus* [34] and *Shigella flexneri* [35].

Bacterial outer membrane proteins are the first molecules to contact host cells, hence they were targeted to be promising vaccine candidates against the invading pathogen [36]. Additionally, they have a significant role in adhesion, nutrient acquisition and maintaining bacterial membrane integrity, therefore they are essential proteins for bacterial life and pathogenesis [37]. Lipopolysaccharide is a major structural component in most Gram-negative bacteria. It makes the outer membrane a barrier to the entry of many antibiotics [38] and, consequently, proteins that have a role in the assembly of lipopolysaccharide (vaccine candidates of the current study) have a high virulence potential which confirms their nomination to be vaccine candidates. 

While pathogens grow quickly, protein extraction and testing for vaccine candidates on a large scale is a costly and time-consuming process, hence the role of immunoinformatics which is capable of decreasing the time needed and saving resources for the discovery of relevant vaccines through epitope mapping, where the non-antigenic regions of an antigen are neglected and only regions that can stimulate B cell and T cell immunity are selected [39]. This approach has been applied successfully and demonstrated promising results when these epitopes were tested on animal models. Actually, this approach has been applied to *M. catarrhalis* where 44 epitopes were predicted, and the top three epitopes were tested and showed superior activity in clearing *M. catarrhalis* from mouse lungs [40].

In the current study, B and T cell epitope prediction has been applied to filtered proteins (LptD and BamA). Th top epitopes have been selected based on percentile rank, antigenicity score, allergenicity, toxicity, conservation and reaction against a number of alleles to cover a high percentage in terms of population coverage. Top-ranked epitopes have been assembled together using appropriate linkers. In addition to that, the beta defensin adjuvant and PADRE peptide were also incorporated into the final construct of the multitope vaccine to strengthen the stimulated immune response and reduce the HLA polymorphism in the population [41].

Validation of the constructed vaccine has been assessed computationally. The constructed vaccine was analyzed according to its physicochemical, structural and immunological characteristics. It was found to be stable, hydrophilic, soluble, antigenic and non-allergenic and its structure validation using Ramachandran plots and ProSA proved that a high-quality 3D structure was modeled. Molecular docking of the designed vaccine with TLR2 showed a low energy score, which proves good binding and high affinity. Molecular dynamics simulation revealed that docked proteins have low deformability, which show the validity of our in silico-predicted vaccine.

## 5. Conclusions

Depending on modern in silico approaches for vaccine development, before biological experiments, is a successful technique that can guide experiments with a high probability of discovering an efficient vaccine with fewer trials, which is a great economic solution. In the current study, reverse vaccinology has been applied to the proteome of 1615 proteins of *M. catarrhalis* and generated two proteins (LptD and BamA) as potential vaccine candidates. Immunoinformatics along with structural vaccinology have been applied to these proteins to design a multitope vaccine, which was validated via online tools to be an efficient vaccine against *M. catarrhalis*. We recommend movement of the constructed vaccine to the biological validation phase using appropriate model organisms to validate our findings.

## Figures and Tables

**Figure 1 vaccines-09-00669-f001:**
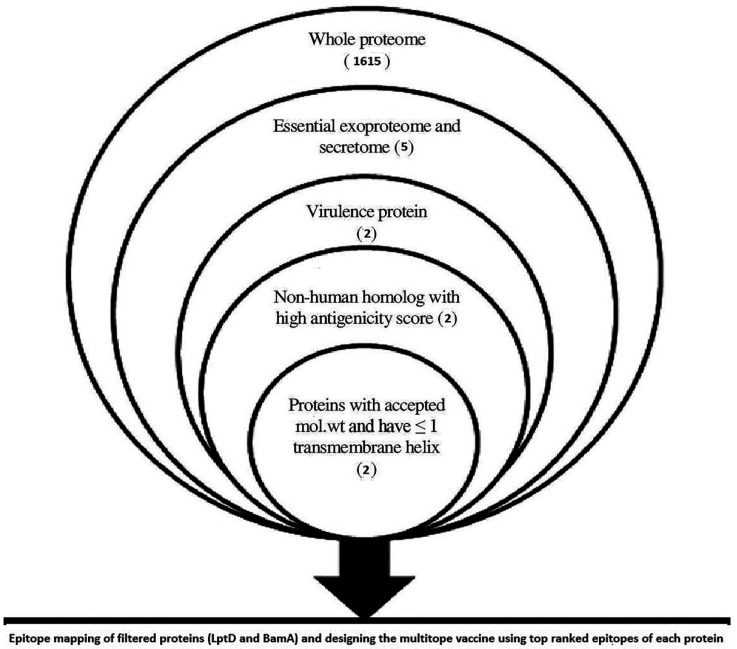
Overview of applied filtration steps for the nomination of potential vaccine candidates by applying reverse vaccinology technique.

**Figure 2 vaccines-09-00669-f002:**
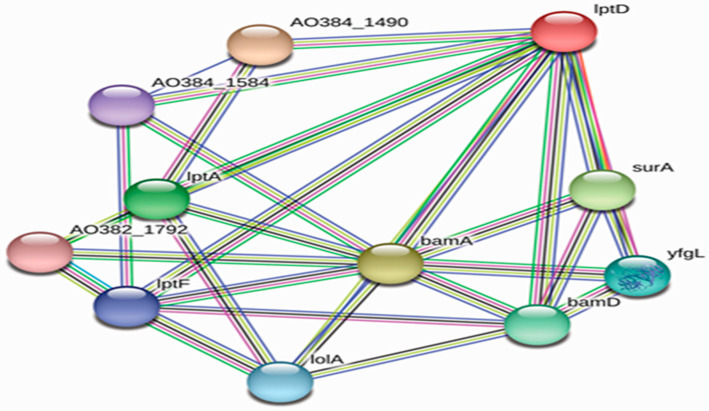
Protein–protein interaction network of vaccine candidates, estimated by STRING database.

**Figure 3 vaccines-09-00669-f003:**
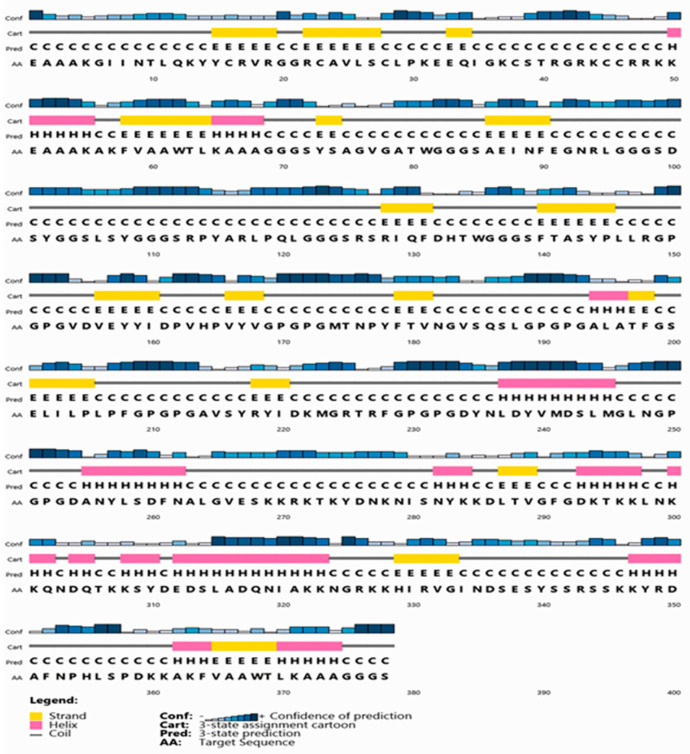
Secondary structure prediction of designed multitope vaccine using PESIPRED server.

**Figure 4 vaccines-09-00669-f004:**
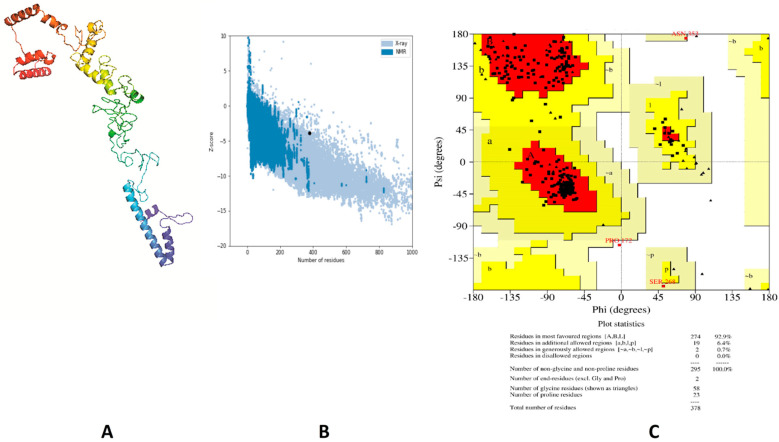
Structural analysis of the designed vaccine. (**A**) The three-dimensional structure of the vaccine obtained after molecular refinements; (**B**) ProSA-web evaluation of the vaccine structure; (**C**) Ramachandran plot analysis of the protein structure after molecular refinements.

**Figure 5 vaccines-09-00669-f005:**
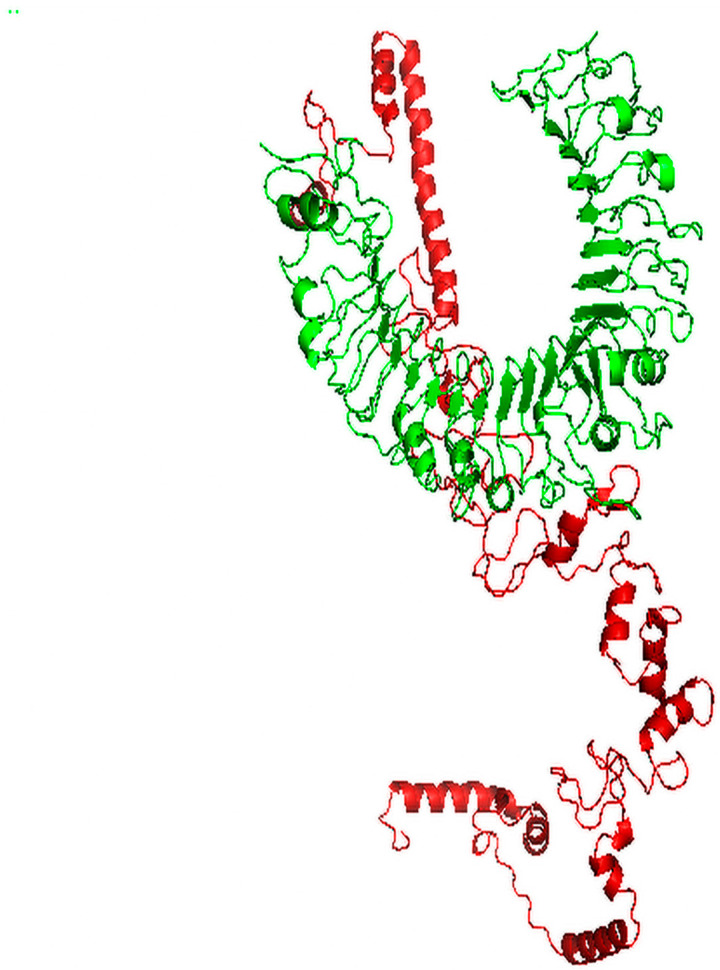
Docked complex of vaccine construct (red) with human TLR2 (green).

**Figure 6 vaccines-09-00669-f006:**
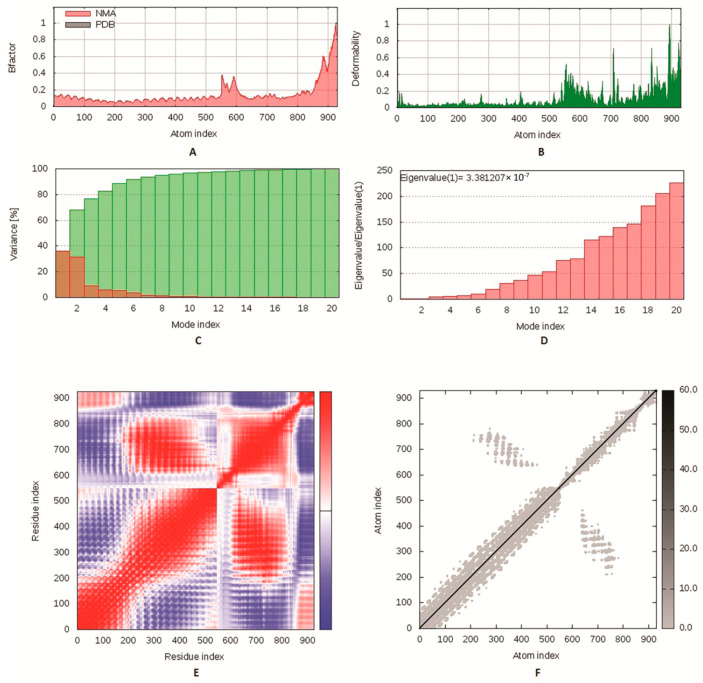
Molecular dynamics simulation of multitope vaccine–TLR2 complex; stability of the protein–protein complex was investigated through B-factor values (**A**), deformability (**B**), variance (**C**), eigenvalue (**D**), covariance of residue index (**E**) and elastic network (**F**) analysis.

**Table 1 vaccines-09-00669-t001:** Characteristics of potential vaccine candidates for *M. catarrhalis* BBH18.

Protein Name	PSORb Result	Essential	Virulent	Non-Human Homolog	Molecular Weight (KDa)	Transmembrane Helices	Antigenicity Score
LptD	Outer membrane	√	√	√	104	1	0.59
BamA	Outer membrane	√	√	√	90.9	0	0.52

**Table 2 vaccines-09-00669-t002:** Top-ranked T cell epitopes (MHC-I peptides) of BamA and LptD proteins.

BamA				LptD			
Epitope	Antigenicity	Allergenicity	Toxicity	Epitope	Antigenicity	Allergenicity	Toxicity
YSAGVGATW	1.16	None	None	AELSGNVIM	0.62	Allergen	None
GEVVGGNAL	1.42	None	None	FEISTPYYL	1.29	Allergen	None
LTQDKQLRY	0.73	Allergen	None	RPYARLPQL	0.41	None	None
RYSAGVGATW	1.31	Allergen	None	RSRIQFDHTW	0.49	None	None
SETREVYSL	0.78	Allergen	None	SEYRLQHVM	0.71	Allergen	None
AEINFEGNRL	0.95	None	None	YEQLLNNNW	0.48	Allergen	None
VQFQIGSVF	0.6	None	None	SSRSSGLAW	0.78	None	None
AEEGFSQAM	0.58	None	None	SYEQLLNNNW	0.41	Allergen	None
IETELTNQY	0.93	Allergen	None	VFYLPYFNF	2.99	Allergen	None
DSYGGSLSY	1.34	None	None	FTASYPLLR	0.93	None	None

**Table 3 vaccines-09-00669-t003:** Top-ranked T cell epitopes (MHC-II peptides) of BamA and LptD proteins.

BamA				LptD			
Epitope	Antigenicity	Allergenicity	Toxicity	Epitope	Antigenicity	Allergenicity	Toxicity
VDVEYYIDPVHPVYV	0.63	None	None	DTGRAIAKNTTLRIK	0.5	Allergen	None
DVEYYIDPVHPVYVR	0.56	Allergen	None	GRAIAKNTTLRIKKV	0.4	None	None
TVDVEYYIDPVHPVY	0.44	None	None	TGRAIAKNTTLRIKK	0.43	Allergen	None
VEYYIDPVHPVYVRR	0.41	None	None	STPYYLNLAPNYDAT	0.91	Allergen	None
TNPYFTVNGVSQSLS	0.77	Allergen	None	TPYYLNLAPNYDATI	0.98	Allergen	None
NPYFTVNGVSQSLSG	1.02	Allergen	None	RIKKVPVFYLPYFNF	1.86	Allergen	None
PYFTVNGVSQSLSGY	0.92	Allergen	None	VSYRYIDKMGRTRFE	0.56	Allergen	None
MTNPYFTVNGVSQSL	0.75	None	None	AVSYRYIDKMGRTRF	0.45	None	None
RPLLTQDKQLRYSAG	0.6	Allergen	None	DYNLDYVMDSLMGLN	0.51	None	None
ALATFGSELILPLPF	0.99	None	None	DANYLSDFNALGVES	0.46	None	None

**Table 4 vaccines-09-00669-t004:** Predicted B cell epitopes of BamA and LptD proteins.

BamA				LptD			
Epitope	Antigenicity	Allergenicity	Toxicity	Epitope	Antigenicity	Allergenicity	Toxicity
TGNFKTQDEV	1.06	Allergen	None	DGGASDHSAGI	1.82	Allergen	None
RREMRQLEGALASNQKIQ	0.6	None	None	KDQQYHDKD	0.81	None	None
RKTKYDNKNISNY	0.7	None	None	KKSIKDNSEPEKSG	0.96	Allergen	None
DLTVGFGDKT	0.85	None	None	SYDEDSLADQNIAKKNGR	0.96	None	None
LNKKQNDQT	1.53	None	None	APFGMHQDT	0.42	Allergen	None
				HIRVGINDSESYSSRSS	1.32	None	None
				RKENRAFNQSAL	0.43	None	None
				YDYNLDYVMDSLM	0.62	None	None
				YRDAFNPHLSPD	1.09	None	None

**Table 5 vaccines-09-00669-t005:** Physicochemical characteristics of the designed multitope vaccine.

Physicochemical Characteristic	Molecular Weight	Theoretical pI	Extinction Coefficient	GRAVY	Instability Index	Aliphatic Index
Score	40.2 kDa	9.71	53,290 M^−1^ cm^−1^	−0.625	37.3	59.68

## Data Availability

The data presented in this study are available on request from the corresponding author.

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
