# Peer review of "In Silico Prediction of a Multitope Vaccine against Moraxella catarrhalis: Reverse Vaccinology and Immunoinformatics"

_vaccines, 2021, doi:10.3390/vaccines9060669_

Round 1
Reviewer 1 Report
It's an interesting manuscript describing an approach to identify potential vaccine candidates. The manuscript needs some improvements, but my primary concern is that the authors did not prove their hypothesis, the production of the recombinant protein, and immunogenicity evaluation. Once the authors have produced the protein, a simple experiment showing the antibody induction will be enough. For the moment, my recommendation is to reject.
Author Response
Reviewer 1:
It's an interesting manuscript describing an approach to identify potential vaccine candidates. The manuscript needs some improvements, but my primary concern is that the authors did not prove their hypothesis, the production of the recombinant protein, and immunogenicity evaluation. Once the authors have produced the protein, a simple experiment showing the antibody induction will be enough. For the moment, my recommendation is to reject.
Answer: The authors thank the reviewer for his intention to improve the impact of our study, however, we need to clarify some points regarding his concern. First, actually we mentioned in the last sentence of the abstract that (additional translational research is required to confirm the effectiveness of the designed vaccine), but at the same time we need to confirm that this is an in-silico approach (as mentioned in the paper title). Second, we intended to further confirm the immunogenicity of the designed vaccine in details through in vivo and in vitro techniques in another research article which we already planned as a research group. Furthermore, many research articles based only on in-silico approaches were published previously in many respectable and high impact journals including (Vaccines) itself. You can check the following published articles: https://doi.org/10.3390/vaccines8030355
https://doi.org/10.3390/vaccines7030088
https://doi.org/10.1016/j.vaccine.2019.04.096
https://doi.org/10.1038/s41598-020-79645-9
https://doi.org/10.1080/07391102.2019.1647286
https://doi.org/10.1016/j.jbi.2019.103160
https://doi.org/10.1016/j.meegid.2019.06.006
https://doi.org/10.1016/j.meegid.2017.08.002
https://doi.org/10.1038/s41598-018-19456-1
https://doi.org/10.1038/s41598-017-09199-w
Reviewer 2 Report
Sltan et al. designed a multitope vaccine against Moraxella catarrhalis (M. catarrhalis) using in silico analysis. They first analyzed the whole proteome of M. catarrhalis, by screening 1615 proteins using reverse vaccinology method, they found outer membrane protein assembly factor BamA and LPS-assembly protein LptD as potential vaccine candidates. Based on these two candidates, they constructed multitope vaccine, by incorporating beta defensin adjuvant and PADRE peptide. The multitope vaccine is 40.2 kDa in size, found to be antigenic, soluble, and stable. It also has a high affinity to its target receptor. Although the author did acknowledge the limitation of in silico study, this study would contribute to the vaccine development field.
Overall, the manuscript is well organized. One impression of this manuscript is that some part of the methods could be moved to results. For example, the authors looked at 1615 proteins, after several steps of analysis, they finally narrowed down the candidates to two. They could describe the flow chart in the Results session, or even they could include a brief flow chart of how they got the BamA and LptD as the best candidates for the vaccine design.
I do not know what kind of software they used for making figures, all letters in the figures were too small, some of them were impossible to read. In addition, the resolution of the figures, especially protein structures need to be fixed.
Minor comments:
- Please introduce the full name of CTL, HTL, BCL, and PADRE in the abstract.
- Table 2-4: Toxicity, I do understand the authors use in silico analysis to predict antigenicity and allergenicity, however, I do not feel comfortable to see even toxicity they can “predict”. I would remove toxicity from the table if the authors do not have a better explanation.
- As mentioned in the general comments, they authors might want to move some methodology to Results.
Author Response
Reviewer 2:
Sltan et al. designed a multitope vaccine against Moraxella catarrhalis (M. catarrhalis) using in silico analysis. They first analyzed the whole proteome of M. catarrhalis, by screening 1615 proteins using reverse vaccinology method, they found outer membrane protein assembly factor BamA and LPS-assembly protein LptD as potential vaccine candidates. Based on these two candidates, they constructed multitope vaccine, by incorporating beta defensin adjuvant and PADRE peptide. The multitope vaccine is 40.2 kDa in size, found to be antigenic, soluble, and stable. It also has a high affinity to its target receptor. Although the author did acknowledge the limitation of in silico study, this study would contribute to the vaccine development field.
Answer: The authors thank the reviewer for his positive impact concerning our study.
Overall, the manuscript is well organized. One impression of this manuscript is that some part of the methods could be moved to results. For example, the authors looked at 1615 proteins, after several steps of analysis, they finally narrowed down the candidates to two. They could describe the flow chart in the Results session, or even they could include a brief flow chart of how they got the BamA and LptD as the best candidates for the vaccine design.
Answer: Ok, the manuscript was modified according to the recommendations Reviewer 2.
I do not know what kind of software they used for making figures, all letters in the figures were too small, some of them were impossible to read. In addition, the resolution of the figures, especially protein structures need to be fixed.
Answer: Thank you for your valuable observation, figures have been produced by photoshop in the format and resolution recommended by the journal in the (instructions for authors) section. If Reviewer 2 or the Editor prefer specific Size or resolution of graphs, we will modify it.
Minor comments:
- Please introduce the full name of CTL, HTL, BCL, and PADRE in the abstract.
Answer: Ok, the full names of CTL, HTL, BCL, and PADRE in the abstract were added as requested.
- Table 2-4: Toxicity, I do understand the authors use in silico analysis to predict antigenicity and allergenicity, however, I do not feel comfortable to see even toxicity they can “predict”. I would remove toxicity from the table if the authors do not have a better explanation.
Answer: The authors think that also toxicity can be predicted through computational tools. Please, kindly check the article (https://doi.org/10.1371/journal.pone.0073957) which explains the methodology of how ToxinPred server -that we used in our study- works. Especially one of the problems that face peptide/protein-based therapy is their toxicity so we think toxicity measurement is necessary.
- As mentioned in the general comments, they authors might want to move some methodology to Results.
Answer: Ok, some methodology was moved to Results as requested.
Round 2
Reviewer 1 Report
Thank you to the authors for the response.
I have a minor observation. Line 95. "... localization of the essential genes." change for " ... localization of essential proteins".
Author Response
Reviewer 1: Thank you to the authors for the response.
I have a minor observation. Line 95. "... localization of the essential genes." change for " ... localization of essential proteins".
Reply: Ok, done.